# Spring-Planted Cover Crop Impact on Weed Suppression, Productivity, and Feed Quality of Forage Crops in Northern Kazakhstan

Gani Stybayev [1], Meisam Zargar [2,*], Nurlan Serekpayev [1], Zhenis Zharlygassov [3], Aliya Baitelenova [1], Adilbek Nogaev [1], Nurbolat Mukhanov [1], Mohamed Ibrahim Mohamed Elsergani [4] and Aldaibe Ahmed Abdalbare Abdiee [1]

[1] Department of Plant Production, Faculty of Agronomy, S. Seifullin Kazakh Agrotechnical University, Astana 010000, Kazakhstan; amd33b@gmail.com (A.A.A.)

[2] Department of Agrobiotechnology, Institute of Agriculture, RUDN University, 117198 Moscow, Russia

[3] Department of Agrotechnology, Faculty of Agronomy, Baitursynov Kostanay Regional University, Kostanay 110000, Kazakhstan

[4] Department of Pesticides, Faculty of Agriculture, Mansoura University, Mansoura 35516, Egypt; misergany100@gmail.com

* Correspondence: zargar_m@pfur.ru

**Abstract:** Integrating cover crops into crop rotation could provide options for herbicide-resistant weed control in farming systems. To evaluate the potential effectiveness of spring-planted cover crop oats (*Avena sativa* L.) on weed suppression, productivity, and feed quality of annual forage crops as sole crops and intercrops in order to determine the best agroecological technique, two-year experiments were laid out under arid conditions in the Akmolinsk region in northern Kazakhstan. Three annual forage crops, namely, (Piper) Stapf.-Sudan grass (*Sorghum sudanense*) (control), common millet (*Panicum miliaceum* L.), and Japanese millet (*Echinochloa frumentacea* L.), and three annual intercropping systems, i.e., 50% pea (*Pisum sativum* L.) + 50% barley (*Hordeum vulgare* L.); 40% pea (*Pisum sativum* L.) + 30% (Piper) Stapf.-Sudan grass (*Sorghum Sudanense*) + 30% barley (*Hordeum vulgare* L.); and 50% pea (*Pisum sativum* L.) + 50% (Piper) Stapf.-sudan grass (*Sorghum Sudanense*), as well as the six mentioned treatments with the sole crops and intercrops plus cover crop oats (*Avena sativa* L.) were used. Japanese millet is a promising newly introduced crop in northern Kazakhstan. It was revealed that the cover crop significantly reduced weed density in the forage sole crops and their intercrops. In all cases, integrating the cover crop with annual forage crops showed higher quality and productivity than non-covered treatments. A highly productive annual crop grown with and without cover intercrop was Sudan grass. The highest yield among the three intercropping systems was recorded with the intercrop constituting 40% pea + 30% Sudan grass + 30% barley. The crude protein content was higher in biomass from sole crops and intercrops constituting cover crops. The overall view was that the use of oats as a cover crop on sole annual forage crops and their intercrops including methods that could be integrated with chemical and non-chemical methods in the field could be a valuable way to reduce weed pressure and improve quality and productivity during the vegetation period.

**Keywords:** annual fodder crops; cover crop; weed suppression; crop quality and productivity

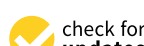



## 1. Introduction

The harsh edaphic and climatic conditions characterized by insufficient heat and moisture during the growing season are limiting factors for year-round forage for animals in northern Kazakhstan. In this regard, there is a need to provide small-holder farms with sufficient good quality forage based on rational management of forage resources depending on soil and climatic conditions. Cover cropping is one of the most promising

strategies to enhance ecological processes in an ecosystem; hence, it is an indispensable component of regenerative agriculture [1] Studies have revealed that annual crops either as sole crops or intercrops show adaptability to the dry conditions in Kazakhstan [2,3]. Annual forage plants are used in particular as green forage in summer; however, they are widely used as intercrops since intercropping systems have been proven to have some advantages over sole crops. They have higher yields of green matter and hay because the plants in intercropping optimally use moisture, light, and nutrients and are less affected by diseases, pests, and weeds [4]. The feed has a higher nutrient ratio, is richer in minerals, and has better palatability and higher digestibility of organic matter, thus being more adaptable for livestock feeding [3–5]. Several studies have confirmed that the use of cover crops is an effective way of controlling weeds, increasing yields, and improving forage quality [6,7]. Weed control by cover crops is mainly performed by depriving weeds of light and other resources during the growth period of the cover crop and through the release of allelochemical compounds into the soil that reduce weed populations by cover-crop residue [2,8–10].

Cover crops are a tool used to control unwanted wild flora but also ultimately an avenue to avoid the use of synthetic chemicals in the soil, preserve microorganisms and biodiversity, and ultimately preserve soil fertility. In many studies, cover crops have been identified as a potential tool for reducing weed population, pests, and diseases, as well as enhancing crop productivity, water retention [11], and improving soil structure [12,13]. In addition, cover crops also improve nutrient cycling, lower leaching [14], and provide winter forage for livestock [15,16]. Obour et al. [7] stated that integrating cover crops during the fallow phase of a crop rotation can significantly control weeds and provide an important control option for herbicide-resistant weeds in farming systems. In Kazakhstan, several studies demonstrated that in the conditions of the steppe and dry steppe zones, where overload increase in pasture, leading to degradation, early spring sowing of perennial grasses leads to stability of the agro-phytocoenoses through the optimization of the processes of restoration of anthropogenically disturbed lands [16].

The perfect choice of a cover crop often diminishes the risk of obtaining low yields and provides the maximum economic efficiency of grass sowing in the area. Previous findings in the south and north-west of Kazakhstan on intercropping the cover crop sweet clover (*Melilotus officinalis* L.) with forage grasses, augmented with various application rates of organic and mineral fertilizers, demonstrated that profitability of the enterprise may reach up to 70% [17]. However, few studies have investigated the impact of cover crops on weeds, productivity, and feed quality of forage crops in Kazakhstan. The objectives of our study were (a) to evaluate the extent of weed control efficacy of spring-planted cover crop oats (*Avena sativa* L.) and (b) to investigate the impact of cover crop on yield and feed quality of annual forage crops as sole crops and in intercropping so as to determine the best cultural technique for arid conditions in northern Kazakhstan.

## 2. Materials and Methods

### 2.1. Site Description and Experimental Design

This study was conducted in two growing seasons, 2020–2021, at the farm "Zerenda" in Tselinograd district, Akmolinsk region, northern Kazakhstan (51°26′1843, 71°09′8232), to investigate the efficacy of spring-planted cover crop oats (*Avena sativa* L.) on weed suppression, productivity, and quality of three annual forage crops, namely, (Piper) Stapf.-Sudan grass (*Sorghum sudanense*) (control), common millet (*Panicum miliaceum* L.), and Japanese millet (*Echinochloa frumentacea* L.), and three intercropping systems as follows: 50% pea (*Pisum sativum* L.) + 50% barley (*Hordeum vulgare* L.); 40% Pea (*Pisum sativum* L.) + 30% (Piper) Stapf.-Sudan grass (*Sorghum sudanense*) + 30% Barley (*Hordeum vulgare* L.); 50% pea (*Pisum sativum* L.) + 50% (Piper) Stapf.-Sudan grass (*Sorghum sudanense*), and also six mentioned treatments with the same crops as sole crops and intercrops plus the cover crop, i.e., oats (*Avena sativa* L.). Randomized complete block design with four replications was used in both experimental years. The blocks comprised of plots measuring 4 m by

30 m (120 m$^2$) consisting of six crop rows (row width of 30 cm). The two central crop rows were used to evaluate and analyze crop yield differences as influenced by the treatments.

Seed sowing was carried out with a grain-grass seeder-SZ-4 ("ASTRA"). The seeding standards given in Table 1 are for sole cropping. The intercrops were sown according to the given percentages of the various components. Immediately after sowing, the soil was rolled with ring-spur rollers 3KKSH-6A to ensure better contact between the sown seeds and the soil. Mowing of the sole crops and intercrops was performed at the beginning of the flowering stage during 10–15 July in both experimental years.

**Table 1.** Sowing depth, sowing date, and sowing rates of forage crops in 2020 and 2021.

| Sowing Characteristics | Sudan Grass | Common Millet | Japanese Millet | Pea | Oats | Barley |
|---|---|---|---|---|---|---|
| Sowing date | 15–18 May | 15–20 May | 15–22 May | 18–22 May | 15–20 May | 18–20 May |
| Depth of sowing (cm) | 6–7 | 6–7 | 6–7 | 5–6 | 6–7 | 6–7 |
| Sowing rate (kg) | 38 | 21.4 | 11 | 180 | 120 | 130 |
| Plant density (plant m$^2$) | 120 | 110 | 120 | 80 | 280 | 300 |

### 2.2. Field Management

During spring, trailed disc harrows (BDM-2.4x2) cut the sod layers and loosened the soil to a depth of 8–10 cm. After using a compact disc harrow, the soil was leveled with the ring-spur roller to prevent it from drying 3KKSH-6A. Basal fertilizer was applied at the recommended rates to all experimental units before planting based on soil test analysis and characteristics. The fertilizer $N_{20}P_{20}K_{20}$ was applied in the experimental plots at a rate of 210 kg ha$^{-1}$, and a top dressing of 120 kg N ha$^{-1}$ was applied when the crops were at the tillering stage. All plants in the experimental plots were irrigated through drip irrigation system distributed along the crop rows. Crop water requirement was calculated by factoring the local evapotranspiration rate of 6.5 mm day$^{-1}$.

### 2.3. Climate Conditions of Survey Area

The study was laid out at a location with an arid climate. The meteorological data during both experimental years 2020–2021 suggest a proper range of temperature, humidity, and precipitation at the experimental field. Precipitation was moderate, with more precipitation in the hot season (six hottest months of the year). The average annual temperature in the region is 3 °C (ranging from −41 °C to +38 °C), and the total annual precipitation varied from 90 to 200 mm. Mean daily temperatures in the spring months (March, April, and May) were −4.1, 5.7, and 5.3 °C higher than the long-term average, respectively. For July and August, the average daily temperatures in the summer were 0.5 and 1.1 °C higher than normal, respectively, and June was at the same level as the annual average. The average daily air temperature of 12.2 °C in September was at the usual level. The precipitation in 2021 fell unevenly: rainfall exceeded the norm by 33.8 and 29.7 mm in the winter months of January and February, respectively (Figure 1).

The precipitation levels in March and May were 10.9 and 21.9 mm below normal, respectively. In April, on the contrary, 11.7 mm more precipitation than the norm was recorded. In the summer months, the maximum precipitation was recorded in June (at the end of the second 10-day period—51.3 mm, at the end of the third 10-day period—42.4 mm) at 94.0 mm and 57.0 mm above the long-term average. In July, precipitation fell by 3.3 mm more than the norm, and in August, by 9.1 mm. In September, rainfall was 6.3 mm higher compared to the long-term average.

Soil characteristics of the experimental field

The soil samples were dried at 65°C, ground, and analyzed with the use of standard methods by Clemson University Agricultural Service Laboratory (Clemson, SC, USA). The soil of the experimental field was classified as loamy (fine-loamy, thermic Typic Kandiudults) with a pH of 7.5 and organic matter 2.3%. Soil samples were taken from two different layers—20 cm and 20 to 40 cm. The soil was typical of the steppe zone of North

Kazakhstan, with low humus and nitrogen, mobile phosphorus and sulfur content, high exchangeable potassium content, and relatively low fertility (Table 2).

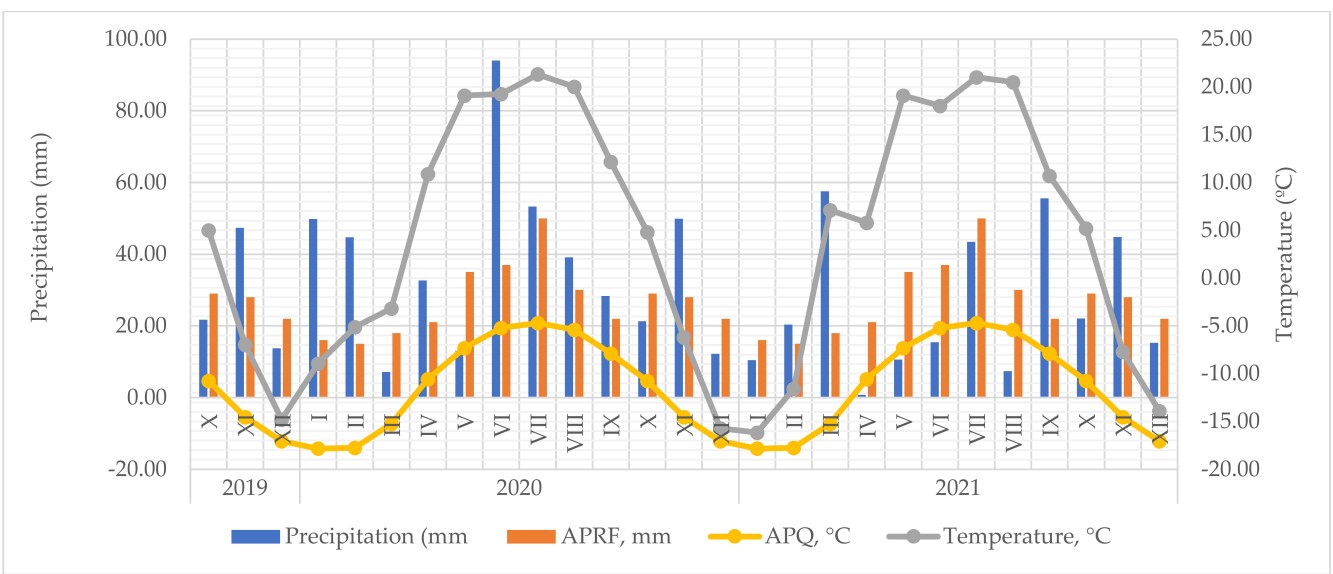

**Figure 1.** Meteorological data during the experimental period (https://www.kazhydromet.kz). Abbreviations: APRF, long-term average rainfall; APQ, average long-term temperature. (accessed on 11 December 2022).

**Table 2.** Soil characteristics at the experimental site.

| Layers, cm | Organic Matter, % | Nitrogen, mg/kg | Phosphorus, mg/kg | Potassium, mg/kg | Sulfur, mg/kg | pH |
|---|---|---|---|---|---|---|
| 0–20 | 2.78 | 8.87 | 24.86 | 614.61 | 4.19 | 7.4 |
| 20–40 | 2.35 | 8.09 | 11.70 | 429.83 | 6.06 | 7.5 |

*2.4. Data Recording*

Each year, weed control was evaluated one week after the tillering stage (during 10–15 June). Weed data from two 0.25 m$^2$ quadrats in each plot were randomly taken from two sampling areas (2 rows by 1 m) in each experimental plot.

The plant density per meter quadrat was determined for each crop; two 0.25 m$^2$ quadrats in each experimental plot were randomly taken during the full shoot stage and before harvesting. The fresh yield of the crops was determined at maturity stage (10–15 July) in both years of the experiments by harvesting three central rows. Immediately after cutting, the green mass was weighed.

Field germination of seeds was determined by counting the number of germinated plants at the time of full germination and the ratio of this number to the number of seeds sown in one square meter in four replicates.

Hay samples were collected for chemical analysis before mowing the herbage. In both seasons, the forage quality was determined on the samples before harvesting. The green mass was crushed and mixed to obtain a uniform sample. The proportion of the individual crop components of the intercropping treatments were determined after harvesting, and on this basis, a sample was made for chemical analysis after shredding and drying. The chemical parameters were determined according to the comprehensive sampling and sample preparation analytical techniques by Michałowski et al. [18]

*2.5. Statistical Analysis*

Before analysis, pooled data were tested for normality and homogeneity of variances. Experiments were carried out in a randomized complete block design (RCBD). All statistical

analysis was conducted by using the SAS and MSTAT-C statistical programs. One-way analysis of variance (ANOVA) was performed on the experimental data. Mean comparisons were computed by the least significant difference (LSD) test. Differences at $p \leq 0.05$ were designated as significant.

## 3. Results and Discussion

### 3.1. Field Germination of Seeds and Plant Density before Harvesting

The interactions of the experimental year and treatments were not significant for the measured traits. According to the ANOVA results, there were significant differences between the annual forage crops and intercrop treatments in integration with and without cover crop oats for both experimental years.

Determination of the plant density at full shoot stage and before harvest indicated that the plant density of the sole crops and intercrops statistically increased in comparison with the control (Sudan grass), ranging from 96 to 145 and 63 to 103 plant/m$^2$, respectively.

The highest plant density at full shoot stage was attained for Japanese millet (145/m$^2$) with cover crop and pea + Sudan grass with and without cover crop (136 and 133/m$^2$, respectively) (Table 3). Assessment of plant density before harvest indicated that the greatest plant density with the value of 104/m$^2$ was observed for Japanese millet without cover crop. On the other hand, the mixture of pea + barley in integration with cover crop had the highest plant density before harvest with the value of 103/m$^2$ average for both experimental years.

**Table 3.** Spring planted cover crop influence on field seed germination and plant density at full shoot stage and before harvesting in 2020–2021.

| Variables | Plant Density in Full Shoots Stage | | Seed Germination | | Plant Density before Harvest | |
|---|---|---|---|---|---|---|
| | Plants m$^2$ | +/− Control | % | +/− Control | Plants m$^2$ | +/− Control |
| *Without cover crop* | | | | | | |
| Sudan grass (control) | 69 ± 4.1 d | - | 69.1 ± 3.2 ab | - | 60 ± 3.4 c | - |
| Common millet | 101 ± 5.2 c | 32 | 50.6 ± 3.9 d | −18.5 | 82 ± 5.5 b | 22 |
| Japanese millet | 145 ± 8.6 a | 76 | 72.5 ± 2.8 a | 3.4 | 104 ± 6.7 a | 43 |
| Pea + barley | 127 ± 6.6 b | - | 63.5 ± 3.1 c | - | 68 ± 3.3 c | - |
| Pea + Sudan grass + barley | 106 ± 7.5 c | −21 | 52.9 ± 4.4 d | −10.6 | 63 ± 3.6 c | 13 |
| Pea + Sudan grass | 136 ± 5.8 ab | 9 | 68.1 ± 3.7 b | 4.6 | 65 ± 4.1 c | -4 |
| *p*-value | 0.0211 | - | 0.028 | - | 0.008 | - |
| Coefficient of variation (%) | 8.05 | - | 5.89 | 4.64 | 8.91 | - |
| *With cover crop* | | | | | | |
| Sudan grass (control) | 90 ± 4.0 cd | - | 60.0 ± 2.5 c | - | 53 ± 2.2 c | - |
| Common millet | 97 ± 3.8 c | 7 | 68.5 ± 1.8 b | 8.5 | 66 ± 2.8 b | 13 |
| Japanese millet | 96 ± 2.8 c | 0 | 60.0 ± 1.9 c | 0 | 67 ± 3.0 b | 4 |
| Pea + barley | 120 ± 6.6 ab | - | 60.0 ± 3.2 c | - | 102 ± 4.8 a | - |
| Pea + Sudan grass + barley | 125 ± 5.7 ab | 5 | 72.0 ± 3.7 a | 12 | 98 ± 4.0 a | − 4 |
| Pea + Sudan grass | 133 ± 7.7 a | 13 | 44.4 ± 0.9 d | −15.6 | 72 ± 2.7 | −30 |
| *p*-value | 0.0025 | - | 0.020 | - | 0.0105 | - |
| Coefficient of variation (%) | 9.09 | - | 3.55 | - | 6.81 | - |

Means within columns followed by different letters are significantly different in terms of Tukey's adjusted means comparisons at $p \leq 0.05$. +/− control, plant density enhancement over control. Increase %, enhancement percentage of plant density over control.

Data analysis on seed germination percentage indicated that the highest increase of 12% was achieved for the intercropping system of pea + Sudan grass + barley in integration with the cover crop compared to the control (Table 3). In congruence with our findings, previous studies have also exhibited the beneficial effects of cover crops. Holman et al. [19] and Nielsen et al. [20] revealed that oats alone or oats intercropped with pea or canola (*Brassica napus* L.) provided favorable weed control and had no negative impact on subsequent forage crop yield when supplemental irrigation was provided. Weil and Kremen [21] reported that using forage radish resulted in improved soybean growth and higher soybean yields.

### 3.2. Weed Suppression

Integrating cover crops in crop rotations can suppress weeds and provide a suitable weed management option for herbicide-resistant weeds in farming systems. In general, the dominant weeds in our experimental fields were wild oat (*Avena fatua* L.) as an annual weed and field bindweed (*Convolvulus arvensis* L.) as a perennial weed, which were observed across the treatments, but some other weed species were also in the field, and they were removed by hand-weeding throughout the trial period.

According to the data analysis, integrating the cover crop oats (*Avena sativa* L.) with annual forage crops in 2020 and 2021 resulted in statistically significant weed control (Table 4).

**Table 4.** Spring planted cover crop effect on annual and perennial weed density in 2020 and 2021.

| Variables | Total Weed Density | | Wild Oat | | Field Bindweed | |
|---|---|---|---|---|---|---|
| | Plants m$^2$ | +/− Control | Plants m$^2$ | +/− Control | Plants m$^2$ | +/− Control |
| Without cover crop | | | | | | |
| Sudan grass (control) | 28 ± 1.2 c | - | 16 ± 1.0 d | - | 12 ± 1.1 b | - |
| Common millet | 39 ± 0.9 b | 11.0 | 24 ± 0.6 b | 8.0 | 15 ± 2.2 a | 3.0 |
| Japanese millet | 45 ± 2.2 a | 17.0 | 31 ± 1.6 a | 15.0 | 14 ± 0.9 a | 2.0 |
| Pea + barley | 24 ± 0.8 cd | −5 | 13 ± 0.3 de | -3.0 | 11 ± 0.5 b | −1.0 |
| Pea + Sudan grass + barley | 18 ± 1.4 d | −10.0 | 18 ± 2.6 c | 2.0 | 7 ± 0.7 d | −5 |
| Pea + Sudan grass | 21 ± 2.1 d | −7.0 | 21 ± 3.0 bc | 5.0 | 9 ± 0.8 c | −3 |
| *p*-value | 0.052 | - | 0.020 | - | 0.022 | - |
| Coefficient of variation (%) | 6.85 | - | 4.99 | 2.10 | 2.00 | - |
| With cover crop | | | | | | |
| Sudan grass (control) | 9 ± 0.5 c | - | 7 ± 1.1 d | - | 2 ± 0.4 c | - |
| Common millet | 18 ± 3.8 b | 9.0 | 13 ± 2.5 b | 6.0 | 5 ± 0.9 a | 3.0 |
| Japanese millet | 28 ± 4.2 a | 19.0 | 24 ± 2.6 a | 17.0 | 4 ± 0.2 ab | 2.0 |
| Pea + barley | 6 ± 0.4 d | −3 | 5 ± 0.3 e | -2.0 | 1 ± 0.3 d | −1 |
| Pea + Sudan grass + barley | 7 ± 0.9 d | −2.0 | 7 ± 0.7 d | - | 1.5 ± 0.5 d | −0.5 |
| Pea + Sudan grass | 10 ± 1.2 c | 1.0 | 10 ± 1.8 c | 5.0 | 3 ± 0.6 b | 1 |
| *p*-value | 0.0205 | - | 0.0090 | - | 0.0041 | - |
| Coefficient of variation (%) | 7.18 | - | 3.88 | - | 10.05 | - |

Data are presented as mean values ± standard deviation (SD). Means within columns followed by different letters are significantly different in terms of Tukey's adjusted mean comparisons at $p \leq 0.05$. +/− control, weed density reduction.

The highest weed numbers of 45 and 28 plants m$^2$ were observed in the treatment of Japanese millet (*Echinochloa frumentacea* L.) with and without cover crop, respectively, which could have been due to the temperature conditions that curtail the growth and affect the plant development [22]. O'Reilly et al. [23] reported that in temperate climates,

annual cover crops are effective in suppressing winter annual weeds in the fall season. The findings of Osipitan et al. [24] showed that when the main crop was planted one to three weeks after cover crop termination, weed suppression was comparable to chemical or mechanical weed control.

Analysis of the effect of the cover crop on weed suppression in June and July in all cases showed significant weed density reduction with the use of cover crop compared to the non-cover crop treatments during 2020–2021 (Table 4). The lowest wild oat (5–7 plants m$^2$) and field bindweed (1–1.5 plant m$^2$) densities were attained for the intercropping systems consisting of pea + barley, and pea + Sudan grass + barley, respectively. Overall, all treatments in combination with the cover crop significantly reduced the population of wild oats and field bindweed compared to the non-cover crop treatments so that the greatest total weed density was obtained for Japanese millet and common millet without cover crop in the values of 45 and 39 plants m$^2$, respectively. Our findings are in agreement with Petrosino et al. [25], who illustrated that spring-planted triticale intercropped with hairy vetch reduced kochia weed density and biomass by 98% in western Kansas.

The data analysis in our study indicated that wild oat (*Avena fatua* L.) and field bindweed (*Convolvulus arvensis* L.) were the most common weeds, which compete for nutrients with the crops. This leads to moisture depletion in the soil and consequently reduces the forage yield and quality. Field bindweed had the highest density in both intercrops and sole crops and can cause intestinal problems in animals because of alkaloids contained in the leaves [26]. Using a cover crop inhibits the development of weeds and significantly reduces their number per meter square and reduces the weed population favorably [7]. Cover crops can control weeds ecologically, physically, and chemically. Cover crops compete with weeds for different resources such as light, nutrients, and water, while some cover crop species release allelochemicals in the soil that hamper weed growth and development [1,24].

Our results show that the cover crop significantly affects weed vegetation in both sole and intercrops, and these results agree with previous findings of other authors who reported that cover crops have a high competitive ability against weeds and can be used as an effective weed control tool in farming systems [27]. Chalise et al. [13] stated that the use of cover crop is beneficial for improving soil properties, conserving soil moisture, and enhancing crop yield.

### 3.3. Green Mass Yields

Cover crops are a promising ecological method to control weeds and enhance crop productivity in farming systems. Data analysis indicated significant differences between the various crops and their mixtures with and without cover crop. The results for green mass yield for the studied growing seasons showed significant differences between the treatments (Table 5). In both years, cover crop out-yielded non-cover-crop treatments because of improved soil properties, moisture, and lower weed density. This is in accordance with the results of Vujic et al. [28], Blanc Canqui and Ruis [29], Toom et al. [30], and Haruna et al. [31]. In 2021, which was somewhat dryer than 2020, yields from all treatments were lower, proving that the meteorological indicators also affected the yields of sole crops and intercropping treatments. In both experimental years, the highest yield was obtained for the intercrop of pea + Sudan grass + barley (16.3–29.2 t ha$^{-1}$) and pea + Sudan grass (21.2–30.4 t ha$^{-1}$), regardless of whether they were grown with or without a cover crop because of the inclusion of drought resistant and high yielding Sudan grass in the intercropping systems [32]. Generally, our results determined that the Sudan grass produced more green mass than the other annuals grown under the arid conditions. The highest average yields of 27.5 and 23.5 t ha$^{-1}$ in 2020 and 2021, respectively, were recorded for the intercrop of pea + Sudan grass and pea + Sudan grass + barley integrating the cover crop. On the other hand, the lowest green mass yield was noted for Japanese millet (10.5 t ha$^{-1}$), while the combination of the cover crop with Japanese millet favorably increased the yield of this crop to an average of 16.4 t ha$^{-1}$ for the two years of the

experiment (Table 5). However, our results over two years showed a desirable efficacy of using cover crops on crop productivity and weed control, which is in accordance with Basche et al. [11], who reported the favorable effect of cover crops on improving yields of subsequent crops.

**Table 5.** Green mass yields of sole crops and intercropping systems with and without the cover crop for 2020 and 2021.

| Variables | 2020 | | 2021 | | Average 2020–2021 |
|---|---|---|---|---|---|
| | Green Mass Yield t ha$^{-1}$ | +/− Control | Green Mass Yield t ha$^{-1}$ | +/− Control | Green Mass Yield t ha$^{-1}$ |
| | | | Without cover crop | | |
| Sudan grass (control) | 18.9 ± 0.2 b | - | 15.0 ± 0.8 c | - | 16.9 |
| Common millet | 15.3 ± 0.5 d | −3.6 | 12.3 ± 1.0 d | −2.7 | 13.8 |
| Japanese millet | 11.1 ± 0.8 e | −7.8 | 10.0 ± 0.7 de | −3.0 | 10.5 |
| Pea + barley | 17.1 ± 2.0 bc | - | 15.1 ± 1.1 c | - | 16.4 |
| Pea + Sudan grass + barley | 27.6 ± 3.2 a | 9.8 | 16.3 ± 2.0 b | 1.2 | 21.9 |
| Pea + Sudan grass | 26.0 ± 3.8 ab | 8.3 | 20.0 ± 2.5 a | 6.1 | 23.0 |
| *p*-value | 0.22 | - | 0.20 | - | - |
| Coefficient of variation (%) | 2.08 | - | 9.01 | - | 7.83 |
| | | | With cover crop | | |
| Sudan grass (control) | 20.2 ± 1.8 bc | - | 18.2 ± 2.4 bc | - | 19.2 |
| Common millet | 17.5 ± 1.4 c | −2.7 | 15.4 ± 1.7 d | −2.8 | 16.4 |
| Japanese millet | 23.3 ± 2.8 b | 2.1 | 12.4 ± 1.8 de | −5.8 | 17.3 |
| Pea + barley | 23.7 ± 2.7 b | - | 19.5 ± 2.0 b | - | 21.6 |
| Pea + Sudan grass + barley | 29.2 ± 3.1 a | 5.5 | 17.8 ± 1.7 c | −1.7 | 23.5 |
| Pea + Sudan grass | 30.4 ± 3.9 a | 6.7 | 24.6 ± 3.2 a | 5.1 | 27.5 |
| *p*-value | 0.020 | - | 0.023 | - | - |
| Coefficient of variation (%) | 6.61 | - | 3.55 | - | - |

Means within columns followed by different letters are significantly different in terms of Tukey's adjusted mean comparisons at $p \leq 0.05$. +/− control, green mass yield enhancement over control.

### 3.4. Chemical Composition of the Hay

The results of the chemical composition of the hay showed that in all cases, both the sole and intercrops with a cover crop had a higher content of dry matter, protein, crude fat, carotene, calcium, and phosphorus. Sudan grass, grown with the cover crop, had the highest dry matter (972.2 g kg$^{-1}$), crude fat (37.6 g kg$^{-1}$), and carotene (29.8 g kg$^{-1}$). The highest protein content (115.1 g kg$^{-1}$) was obtained in Japanese millet + pea + barley with the cover crop, and common millet had the lowest protein content 90.7 g kg$^{-1}$. The intercropping system comprising pea + Sudan grass with the use of cover crop in both years had the highest fiber, sugar, and phosphorous with values of 330.3, 109.7, and 4.4 g kg$^{-1}$, respectively. The maximum ash (102 g kg$^{-1}$) and calcium (25.5 g kg$^{-1}$) were observed for common millet and the combination of pea + Sudan grass + barley with the cover crop (Table 6).

**Table 6.** Spring-planted cover crop effects on dry matter and chemical composition of the hay in 2020–2021.

| Variables | Dry Matter g kg⁻¹ | Protein | Fat | Fiber | Ash | Sugar | Carotene | Ca | p |
|---|---|---|---|---|---|---|---|---|---|
| | Chemical Composition (g kg⁻¹) | | | | | | | | |
| | | | | Without cover crop | | | | | |
| Sudan grass (control) | 952.2 a | 101.7 a | 27.5 a | 304.9 ab | 88.3 b | 45.1 d | 19.4 a | 9.7 d | 2.3 c |
| Common millet | 947.3 a | 90.7 b | 27.6 a | 303.2 ab | 92.5 a | 32.3 e | 19.85 a | 11.1 c | 2.3 c |
| Japanese millet | 794.7 c | 92.0 b | 24.2 ab | 265.8 c | 70.9 c | 83.2 bc | 16.7 b | 11.5 c | 3.2 a |
| Pea + barley (Control) | 925.3 ab | 105.9 a | 20.5 b | 306.3 ab | 87.3 b | 91.7 ab | 15.5 b | 15.3 a | 3.1 ab |
| Pea + Sudan grass + barley | 921.6 ab | 96.4 ab | 15.2 c | 319.7 a | 82.6 bc | 87.1 b | 11.3 c | 15.5 a | 3.2 a |
| Pea + Sudan grass | 930.4 a | 104.0 a | 24.0 a | 320.3 a | 70.3 c | 99.8 a | 14.8 b | 14.2 b | 3.4 a |
| *p*-value | 1.051 | 0.021 | 1.022 | 0.022 | 0.001 | 1.025 | 0.051 | 0.009 | 1.0 |
| Coefficient of variation (%) | 6.25 | 2.05 | 6.60 | 4.52 | 1.08 | 7.75 | 5.18 | 8.19 | 3.3 |
| | | | | With cover crop | | | | | |
| Sudan grass (control) | 972.2 a | 111.7 a | 37.6 a | 314.9 b | 98.3 a | 55.1 d | 29.8 a | 19.7 cd | 3.3 b |
| Common millet | 953.3 ab | 96.2 b | 37.4 a | 313.2 b | 102.0 a | 42.3 e | 29.3 a | 21.0 c | 3.3 b |
| Japanese millet | 824.7 c | 102.0 ab | 34.2 ab | 275.8 c | 80.9 c | 93.2 bc | 26.7 ab | 21.5 c | 4.2 a |
| Pea + barley | 945.3 ab | 115.1 a | 30.5 b | 316.3 b | 97.3 ab | 101.7 ab | 25.5 b | 25.0 a | 4.1 a |
| Pea + Sudan grass + barley | 941.6 ab | 106.4 a | 25.2 c | 329.7 a | 92.6 b | 97.1 b | 21.3 c | 25.5 a | 4.2 a |
| Pea + Sudan grass | 940.4 ab | 114.0 a | 34.0 ab | 330.3 a | 80.3 c | 109.7 a | 24.8 bc | 24.2 b | 4.4 a |
| *p*-value | 0.020 | 0.020 | 0.009 | 0.007 | 0.018 | 0.010 | 0.087 | 0.069 | 0.001 |
| Coefficient of variation (%) | 10.28 | 2.08 | 6.17 | 8.11 | 6.19 | 9.27 | 4.37 | 3.88 | 7.64 |

Means within columns followed by different letters are significantly different in terms of Tukey's adjusted mean comparisons at $p \leq 0.05$.

In agreement with the results of preceding authors, our findings indicated that Sudan grass is a source of energy and protein, has high nutritive value, and is beneficial to the improvement of forage palatability and digestibility [33]. Using a high-protein crop pea in intercrops enriches the forage with protein and other essential elements, as postulated by Vasin et al. [34]. The forage obtained from sole crops and the intercropping systems, grown with the cover crop, had a higher protein, ash, carotene, calcium, and phosphorus content because of reduced weed infestation, which tallies with the findings of Hartwig and Ammon [35] who demonstrated the efficacy of cover-cropping on reducing weed populations as well as enhancing crop productivity. However, Vujic et al. [28] reported that the cover crop benefits are not present in energy or forage systems where biomass is harvested, while Drewnoski et al. [36] stated that using cover crops as a forage source can provide the opportunity for short-term economic and soil conservation benefits for the farmers and agricultural systems. When climatic conditions are not a limiting factor, the use of cover crops positively impacts forage production, having in mind that desirable production of both crops would have been obtained as well as the secure and more diversified crop cultivation [37–39]. Hence, introducing cover crops would significantly contribute to enhancing the sustainability of the existing agricultural systems across the globe.

## 4. Conclusions

The results of the two-year experiment conducted in northern Kazakhstanin Akmolinsk region show that cover crops had a positive effect on weed suppression and crop productivity. The study demonstrated that cover crops reduced weed density whilst simultaneously increasing yield and the quality of annual sole fodder crops and their intercrops.

The highlights were that using oats as a cover crop and integrating them with annual fodder crops and their intercrops might be a valuable cultural strategy to reduce weed pressure as well as increase the quality and productivity of crops under dry conditions, even as a method that could be integrated with other chemical and non-chemical methods. Integrating a cover crop with annual forage crops indicated higher quality and productivity than non-covered treatments. A highly productive annual crop grown with and without cover crop was Sudan grass. Generally, using a cover crop is an agroecological strategy to control weeds and is also ultimately a tool to avoid the use of synthetic chemicals in the soil and preserve microorganisms and biodiversity, thus leading to sustainable crop production.

**Author Contributions:** G.S. and M.Z.; Conceptualization, Methodology, data analysis, validation & investigation; N.S. and M.I.M.E.; Formal analysis, Writing—original draft; Z.Z.; Writing—review & editing, project administration; A.B.; Conceptualization, Project administration, Resources, Software, Writing—review & editing; A.N. and N.M.; Conceptualization, Writing—original draft, Supervision; Conceptualization, Supervision, Validation, Supervision, Validation, Writing—review & editing, data analysis, validation & investigation, writing original draft preparation; A.A.A.A.; Conceptualization, data analysis, Validation. All authors have read and agreed to the published version of the manuscript.

**Funding:** The work was prepared on the basis of the results of scientific research on the topic of the IRN AR08052781 project "Development of a raw material conveyor for the year-round supply of high-quality feed of MRS (dairy goats) in the conditions of the arid steppe of the Akmola region" under the budget program 217 "Development of science". This paper was also supported by the RUDN University Strategic Academic Leadership Program.

**Data Availability Statement:** The data presented in this study are available in the article.

**Conflicts of Interest:** The authors declare no conflict of interest.

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
