# Peer review of "Spring-Planted Cover Crop Impact on Weed Suppression, Productivity, and Feed Quality of Forage Crops in Northern Kazakhstan"

_agronomy, doi:10.3390/agronomy13051278_

Round 1
Reviewer 1 Report
Dear Authors,
After reviewing the manuscript, I believe that the topic of the research undertaken is very interesting and relevant to agronomy. Carried out are extensive, a lot of hard research work has been done. However, in my opinion, the manuscript needs many additions and clarifications.
Line 48 add reference to literature
Line 93 Were oat as a cover crop sown at the same time as the main crop? This can be inferred from Table 1 and the title of the manuscript. However, was CC sowing at a different time?
Line 99 How were the mixtures seeded? In one pass with the seeder or each component in a separate pass? If so, was the sowing parallel or perpendicular? Was inoculation used for pea seeds?
Line 116 (Table 1) Are the reported seeding rates in kg/ha average values from 2 years of experiments? I suggest adding the plant density when sowing per m2 or ha. Legumes in particular can vary in weight of 1,000 seeds from year to year so giving seeding rates in kg/ha is not very precise. Also, the plant density analyzed further suggests the need to specify sowing seeds per m2. I understand that these are the given seeding standards for pure sowing and according to the percentage of mixture components given earlier, the sowing of mixtures was realized? If so it should be clearly stated.
Line 154 - 156 Delete, a more appropriate place for this sentence is the results section
Line 162 - 163 Earlier it was mentioned that the collection was July 10 - 15. Please clarify this as it is unclear.
Line 165 How were samples uptaken for chemical analysis? I am thinking of mixtures in particular. Was harvesting done, the green mass was crushed and mixed to obtain a uniform sample? Was the proportion of the individual components of the mixtures determined after harvesting, and on this basis, after shredding and drying, a sample was created for chemical analysis. The second case is much more appropriate because when harvesting mixtures, there can be a great deal of randomness when it comes to sampling the experimental plot, due to the possible competitiveness of crops against each other. In particular, legumes such as peas are sensitive to competition from cereal crops.
Line 177 Results and discussion
Line 193 I don't quite understand the point of comparing the stocking rates of different crops (mixtures with different components, pure sowing) with each other. Since different numbers of sown seeds per m2 were used, I suppose, this will automatically translate into different stocking rates. Or perhaps the exact same number of seeds per m2 were sown for each m2, although this seems unlikely to me. It would be more logical to compare the planting density between a given crop with and without CC
Line 203 References to studies by other authors refer to weediness while the paragraph on results about germination. This is inappropriate or why such a reference was used should be stated.
Line 205 It is not clear to me what the control is in this table. What is the +/- control referring to? I suppose in the case of Common millet and Japanese millet to Sudan grass (Control) while mixtures with Sudan grass to Pea + Barley however this is not consistent in all lines. Since the tables show average values for the experiment I suggest adding the standard deviation. In my opinion, there is a problem with comparing the stocking rates of different seedings throughout the table. I would suggest more of a comparison between crops with and without CC.
Line 229 The authors in the table compare weeds between crops and not between a given crop with and without CC (letters in the table)
Line 231 I suggest adding a reference to the literature and explaining why the mixtures reduced weeds
Line 241 This is very valuable information that suggests assessing not only the number of weeds per m2 but also their dry weight. Has there been an assessment of the dry weight of weeds? It is important especially when the crop is harvested for green fodder. If so, please add this data.
Line 256 (Table 4) For this table, the +/- control column refers to Sudan grass (Control) throughout the table. This is confusing compared to the previous table. Also, I suggest adding the standard deviation. I would suggest rebuilding the table to make it visible and clear to compare weeds in crops with and without CC.
Line 260 If oats as CC were sown together with the proper crop I think a more appropriate approach would be to treat it as a component of mixtures rather than CC. Since it is likely that the green mass of oats was important in increasing the crops proper
Line 289 I suggest giving the DM in t/ha this will be more illustrative of the yields obtained. In the current presentation, it more represents the vegetative state of the crop at harvest rather than the livestock feed obtained. As in previous tables, I suggest adding the standard deviation. In the description of the results, I suggest stating the differences obtained in % or g/kg rather than repeating the results in the tables, I suggest doing this throughout the manuscript. In addition, I suggest describing the mechanisms that caused the differences in chemical composition and not just refer to the studies of other authors. Such as the possible symbiosis of pea plants with bacteria, or different root structure.
Throughout the paper, I would suggest changing the form of the tables for a clearer presentation of the results. I suggest dropping the +/- control column instead of describing this in the results section. (Example for Table 4 attached).
For individual tables:
Table 5. I suggest putting green and dry matter in t/ha in one table.
Table 6. in one table Protein, Fat, Fiber, Ash in the other Sugar, Carotene, Ca, P as suggested for the tables

Author Response
Dear Reviewer
We gratefully acknowledge the detailed revision of the text and useful suggestions to improve the paper by the reviewer. We have closely followed he/she suggestions and introduced the required changes in the text. Main changes are highlighted into the manuscript in YELLOW. Below, we have included reviewer comments and our responses.
- Reference added to the literature in line 47 highlighted in YELLOW.
- According to the suggestion, yes, sowing date and sowing rates of crops in presented in table 1. Oats planted 15 – 20 May in the study.
- Plant density added in the table 1 as plant per meter quadrate.
- Inoculation was not used for pea seeds. Seed sowing was carried out with a grain-grass seeder. Each component performed in a separate pass.
- Table 1 is the reported seeding rates in kg/ha-1 average values for 2 years of experiments 2020 – 2021.
- Plant density parameter added to the table 1 as suggested by the reviewer.
- Line 154 - 156 from M&M and added to the R&D section.
- Line 162 - 163 revised. The Fresh yield mass of the crops was determined at maturity stage (10 - 15 July) in both years of the experiments.
- Chemical analysis part in M&M revised.
- Line 203 revised according to the reviewer suggestion.
- In line 205: +/- control, plant density enhancement over control; Increase %, enhancement percentage of plant density over control. We compered plant density and seed germination of all crops and mixtures with control plots as (+/- control).
- Standard deviation (SD) added to the table 3. Table style revised.
- Author did not understand reviewer comment about line 229.
- A related reference added to the Line 233 explaining why the mixtures reduced weeds as suggested by the reviewer.
- About the comment for line 241: we did not measure dry weight of weeds in this study.
- Style of table 4 revised according to the reviewer suggestion. Also, standard deviation (SD) added to the means.
- Table 5 also revised, and standard deviation (SD) added to the data. Style of the table 5
- Dear reviewer we revised and changed style of all tables. Please check if they are appropriate.
We hope that after these enhancements the manuscript can now be accepted, although we are certainly willing to consider further changes if necessary.
Yours sincerely,
Reviewer 2 Report
Dear Authors,
Congratulations on your work, which I appreciated in its simplicity but also in its completeness of experimental information and adopted design. The manuscript relates to a globally important topic because it represents a useful agroecological transition tool to mitigate the direct or indirect effects of the climate crisis. The control of undesirable flora through the use of cover crops represents a model of sustainability in soil management as an alternative to the use of herbicides (in orchards) or to improve the quality of the flora present in the case of permanent grasslands and pastures. To this end, I give only two suggestions for consideration in the introduction and conclusion of the manuscript:
- Introduce both in Introduction and Conclusions sections the concept of agroecological transition as a tool for sustainability. Cover crops are a tool to control unwanted wild flora but also ultimately a tool to avoid the use of synthetic chemicals in the soil, preserve microorganisms and biodiversity, ultimately to preserve soli fertility.
- Include both in Introduction and Conclusions sections some concepts related to the importance of rational grazing, which is ultimately the only tool to preserve the ecosystem related to the species that are used to feed the animals
Author Response
Dear Reviewer
We gratefully acknowledge the detailed revision of the text and useful suggestions to improve the paper by the reviewer. We have closely followed he/she suggestions and introduced the required changes in the text. Main changes are highlighted into the manuscript in YELLOW. Below, we have included reviewer comments and our responses.
Authors addressed your suggestions in introduction and conclusion sections
We hope that after these enhancements the manuscript can now be accepted, although we are certainly willing to consider further changes if necessary.
Yours sincerely,
Reviewer 3 Report
Abstract: The text needs to be thoroughly improved. There are very long sentences that make the sense of what is intended to be said in the manuscript lost. In general, I do not quite understand the objectives pursued in the ms.
Material and methods: Two growing season are considered appropriate in order to demonstrate the results reached. It could be interesting add a summary table with the treatments in order to be more clear.
The conclusions are too brief, I need more information about the results obtained and how it can be implemented in crop systems. Long-term expectations and recommendations linked with other studies.
Author Response
Dear Reviewer
We gratefully acknowledge the detailed revision of the text and useful suggestions to improve the paper by the reviewer. We have closely followed he/she suggestions and introduced the required changes in the text. Main changes are highlighted into the manuscript in YELLOW. Below, we have included reviewer comments and our responses.
- Abstract of the manuscript revised according to the reviewer suggestion.
- Conclusion revised and added some information.
- Based on your recommendations, we tried to clearly describe experimental treatments in the article, especially in the abstract and also they (experimental treatments) are fully presented in tables 3 – 6.
We hope that after these enhancements the manuscript can now be accepted, although we are certainly willing to consider further changes if necessary.
Yours sincerely,
Round 2
Reviewer 1 Report
Dear Authors,
Thank you for making some of the suggested corrections to the manuscript. However, I believe that the manuscript still needs corrections and clarifications.
Line 103 I suggest supplementing with the sentence, "The seeding standards given in Table 1 are for pure sowing. The mixtures were sown according to the given percentages of the various components." If this was indeed the case, otherwise please clarify
Table 3 The authors state in the explanation: Means within columns followed by different letters are significantly different by Tukey adjusted means comparisons at P ≤ 0.05. However, in some columns the P-value is above 0.05 and yet letters appear saying significant differences. Please explain this. The comment also applies to other tables.
Please clarify how Seed germination was determined, it is not mentioned in the M&M chapter.
I repeat my question regarding the comparison of Plant density in full shoots stage and Plant density before harvest between Variables (Sudan grass (Control), Common millet, Japanese millet, Pea + Barley, Pea + Sudan grass + Barley, Pea + Sudan grass). If, for example, Pea + Barley (80*50% + 300* 50% = 190 seeds/m2; Pea + Sudan grass + Barley ( 80*40% + 120*30% + 300*30% = 158 seeds/m2) Pea + Sudan grass (80*50% + 120* 50% = 100 seeds/m2) so sowed different amounts of seeds per m2 which should translate into the analyzed trait. Wouldn't it be more illustrative to compare individual Variables between the crop with and without CC? For example, Sudan grass Without cover crop and Sudan grass With cover crop? Please address this and explain why the comparisons in the table were chosen.
In addition, several places in the table revealed Plant density in full shoots stage greater than the number of seeds sown. Please address what this could have been due to.
Line 229 - 231 The authors state in the lines: "Analysis of the effect of the cover crop on weed suppression in June and July in all cases showed significant weed density reduction with use of cover crop compared to the non-cover crop treatments during 2020 - 2021." So they suggest significant differences between cultivation with and without CC. However, the comparison of means in Table 4 (Means within columns followed by different letters are significantly different by Tukey adjusted means comparisons at P ≤ 0.05.) compares the influence of Variables: Sudan grass (Control), Common millet, Japanese millet, Pea + Barley, Pea + Sudan grass + Barley, Pea + Sudan grass on weed incidence. In order for the sentence given above to be consistent with the table, one would have to compare, for example: Sudan grass Without cover crop and Sudan grass With cover crop.
The suggestion given applies to the entire manuscript. The title mentions the influence of CC while all tables separately compare crops with and without CC. At least that is what the Tables show, if it is different the Tables should be rebuilt to make it clear.
Table 5 The authors obtained greater green mass yields after cultivation with CC in their study. Since it was stated in the M&M section that the trials were separated into individual components to create samples for chemical analysis, was the presence of oat demonstrated? A possible situation is that the increase in green mass yield was achieved through plant density due to the addition of oat in the seeding.
Yours sincerely
Author Response
Dear Reviewer
We gratefully acknowledge the detailed revision of the text and useful suggestions to improve the paper by the reviewer. We have closely followed he/she suggestions and introduced the required changes in the text. Main changes are highlighted into the manuscript in YELLOW. Below, we have included reviewer comments and our responses.
- Line 103 revised according to the reviewer suggestion.
- Data recording related to seed germination added to the M&M part.
- Unfortunately, we do not understand mentioned comment of the reviewer very well regarding the comparison of Plant density in full shoots stage and Plant density before harvest between Variables. The aim of our study was to evaluate extent of weed control efficacy of CC and investigating impact of CC on yield and feed quality of annual forage crops and their mixtures in order to determine the best cultural technique for arid conditions in northern Kazakhstan. In fact, we sought to determine the best cultural method to solve the relevant problems in the region. And desirable results were achieved in the study.
- Table checked for p values. Our co-author made a typo mistakes in some cases. Thank you for your advice.
- About the question on Line 229 – 231. Authors initially stated that generally weed density reduction with the use of cover crop was better compared to the non-cover crop treatments, then after this sentence we have interpreted and explained the details as we mention following text: The lowest wild oat (5 - 7 plants m2) and field bindweed (1 - 1.5 plant m2) density were attained for mixture of pea + barley, and pea + sudan grass + barley respectively. Totally, all treatments in combination with the cover crop significantly reduced population of wild oats and field bindweed compared to the non-cover crop treatments, so that the greatest total weed density was obtained for Japanese millet and common millet without cover crop in the value of 45 and 39 plants m2
- Title of tables revised.
- Presence of oat was demonstrated.
We hope that after these enhancements the manuscript can now be accepted, although we are certainly willing to consider further changes if necessary.
Yours sincerely,
Reviewer 3 Report
According to the revised ms and taking into account all modifications followed by the authors, I can consider acceptable in the present form the ms to be published in the journal.
Author Response
Dear reviewer
Thanks for accepting our manuscript
Meisam,